# Molecular Dynamics and MM-PBSA Analysis of the SARS-CoV-2 Gamma Variant in Complex with the hACE-2 Receptor

**DOI:** 10.3390/molecules27072370

**Published:** 2022-04-06

**Authors:** Maurizio Cavani, Walter Arnaldo Riofrío, Marcelino Arciniega

**Affiliations:** 1Faculty of Science and Philosophy, Universidad Peruana Cayetano Heredia, Av. Honorio Delgado 430, San Martín de Porres, Lima 15102, Peru; maurizio.cavani.b@upch.pe; 2Department of Statistics, Demography, Humanities and Social Sciences, Faculty of Science and Philosophy, Universidad Peruana Cayetano Heredia, Av. Honorio Delgado 430, San Martín de Porres, Lima 15102, Peru; walter.riofrio.r@upch.pe; 3Department of Biochemistry and Structural Biology, Institute of Cellular Physiology, National Autonomous University of Mexico, Mexico City 04510, Mexico

**Keywords:** SARS-CoV-2, VoC’s, gamma variant, compensatory mutations, molecular dynamics, MM-PBSA, residue specific interaction entropy

## Abstract

The SARS-CoV-2 virus, since its appearance in 2019, has caused millions of cases and deaths. To date, there is no effective treatment or a vaccine that is fully protective. Despite the efforts made by governments and health institutions around the globe to control its propagation, the evolution of the virus has accelerated, diverging into hundreds of variants. However, not all of them are variants of concern (VoC’s). VoC’s have appeared in different regions and throughout the two years of the pandemic they have spread around the world. Specifically, in South America, the gamma variant (previously known as P.1) appeared in early 2021, bringing with it a second wave of infections. This variant contains the N501Y, E484K and K417T mutations in the receptor binding domain (RBD) of the spike protein. Although these mutations have been described experimentally, there is still no clarity regarding their role in the stabilization of the complex with the human angiotensin converting enzyme 2 (hACE-2) receptor. In this article we dissect the influence of mutations on the interaction with the hACE-2 receptor using molecular dynamics and estimations of binding affinity through a screened version of the molecular mechanics Poisson Boltzmann surface area (MM-PBSA) and interaction entropy. Our results indicate that mutations E484K and K417T compensate each other in terms of binding affinity, while the mutation N501Y promotes a more convoluted effect. This effect consists in the adoption of a *cis* configuration in the backbone of residue Y495 within the RBD, which in turn promotes polar interactions with the hACE-2 receptor. These results not only correlate with experimental observations and complement previous knowledge, but also expose new features associated with the specific contribution of concerned mutations. Additionally, we propose a recipe to assess the residue-specific contribution to the interaction entropy.

## 1. Introduction

The pandemic caused by the SARS-CoV-2 virus has hit the world’s population with the loss of millions of human lives. As of March 2022, over 452 million cases have been confirmed around the world, including over 6 million deaths due to severe acute respiratory syndrome [1]. Despite the efforts of local governments to vaccinate the entire population, the appearance of new variants of SARS-CoV-2 has been associated with an increase in infections (in some cases reinfections) and deaths in recent months.

Viruses require cellular machinery to multiply and infect surrounding cells. Viral replication occurs at high speed, allowing errors or mutations to accumulate between generations. These mutations in some cases potentiate the virus, yielding more aggressive and transmissible strains. Particularly with SARS-CoV-2, variants have progressively appeared in different countries, thereby generating individual characteristics, and opening new paths in the evolutionary process. However, only a few of these variants are cataloged as variants of concern (VoC’s) due to their abundance in the infected population and their wide geographic distribution [2]. Protein S or spike protein is the main viral protein responsible for binding to the hACE-2 receptor in host cells. Likewise, it is the protein that is tracked by mutations to denote new variants of the virus [3,4].

In the first two detected SARS-CoV-2 VoC’s B.1.1.7 (alpha) and B.1.351 (beta), multiple retrospective studies have been conducted to understand the impact of mutations in the virus transmissibility and the effectiveness of the vaccines developed throughout 2020 [5]. The alpha variant that emerged in England has been detected in several countries and is associated with increased transmissibility due to the affinity for the hACE-2 receptor [6,7]. It is also associated with a possible evasion of the immune system due to less susceptibility to neutralizing antibodies and greater risk of death [8,9]. The alpha variant has three mutations of interest in the spike protein: (i) the N501Y mutation that corresponds to the receptor-binding motif (RBM); (ii) a 69/70 deletion in the receptor binding domain (RBD) that triggers a notable change in the conformation of the RBM; and (iii) the P681H mutation found near the furin S1/S2 cleavage site. The beta variant, also known as 20H/501Y.V2, was identified in South Africa and is characterized by carrying the K417N, E484K and N501Y mutations in the RBM of the spike protein [10]. Although there is no evidence suggesting that the beta variant presents a different level of aggressiveness with respect to its alpha counterpart, it has been proven that mutations in the spike protein affect the virus neutralization by antibodies [11]. This evidence indicates that the evolutionary process of SARS-CoV-2 since the beginning is directed towards the evasion of the immune system [12,13].

Due to the rapid evolutionary process of SARS-CoV-2, new variants appeared with new combinations of mutations, making them more dangerous and bringing new waves of infection. Among those are variant P.1 (gamma), variant B.1.617.2 (delta) and variant B.1.1.529 (omicron). The Brazilian gamma variant was identified as the cause of the new outbreak that occurred in Manaus and Brazil between November 2020 and January 2021, by the National Institute of Infectious Diseases of Japan during a routine sampling of four travelers from Brazil [14,15]. We decided to study the de gamma variant since, at the time when the manuscript was conceived, this variant was still considered to be a health problem in Latin America; particuarly in South American countries which, throughout early 2021, had the greatest impact [16]. The gamma variant carries the K417T, E484K and N501Y mutations previously described in RBM [17]. As a consequence of these mutations, studies with monoclonal antibodies stablished that the gamma variant possess the ability to be more resistant to neutralization [18,19]. Likewise, the interaction with the human angiotensin converting enzyme 2 (hACE-2) receptor has been experimentally studied. While the K417T mutation results in the loss of a salt bridge with D30, mutation E484K increases the electrostatic complementarity of the binding partners. Additionally, the mutation N501Y exhibits a compensatory effect since it favors π-π interaction with the residue Y41. The overall effect of these three mutations causes a small increment in the K_D_ of the gamma variant towards the hACE-2 receptor [19,20]. Based on this experimental evidence, in this work we build the gamma variant complex and assess the binding energy contribution effect that these mutations, at the RBM of SARS-CoV-2 spike protein, imprint on the interaction with the hACE-2 receptor. These evaluations were carried out by performing molecular dynamic simulations and interaction-free energy analysis.

## 2. Materials and Methods

### 2.1. Molecular Dynamics

The three-dimensional structures of the SARS-CoV-2 wild type receptor binding domain in complex with the receptor hACE-2 (PDB ID: 6M0J) [21] and the gamma variant spike protein receptor-binding domain in complex with COVOX-222 and EY6A Fabs (PDB ID: 7NXB) [19] were obtained from Protein Data Bank [22].

The gamma variant spike protein receptor-binding domain was separated from the antibodies (7NXB complex). In both structures, water molecules, hydrogens, and glycosylation were removed. The known ionic cofactors Zinc and Chlorine were maintained. The gamma variant complex was constructed through a C-α structural alignment between the wild type complex and the gamma variant spike protein receptor-binding domain (RMSD = 0.44 Å) using PyMOL v.2.0 [23].

The preparation of the complexes for simulation by molecular dynamics were done with the GROMACS v.2020.1 [24] employing the Amber99sb-ildn [25]. The simulated protein complexes were placed in a triclinic box with length sizes extending 11 Å from the extremes of the molecule in each dimension. A TIP3P water molecule model was used to solvate the system. Chloride and sodium ions were added to reach a 0.150 M of NaCl, together with a small surplus of ions, to neutralize the system’s electrostatic charge. All the MD simulations were carried out considering periodic boundary conditions. The systems were energy minimized for 50,000 steps using the steepest descent algorithm. The temperature was set at 300 K and employed the Berendsen thermostat during 1ns of simulation, while position restrains of 1000 kJ mol^−1^ nm^−2^ were applied on heavy atoms of the protein. Subsequently, the pressure was equilibrated at 1 bar aided with the Berendsen barostat during 1ns while position restrains of 1000 kJ mol^−1^ nm^−2^ were applied on heavy atoms of the protein’s main chain. A final equilibration step of 1 ns was performed without applying position restrains. During this period the temperature and pressure were maintained using the Verlet thermostat and Parrinello-Rahman barostat, at 300 K and 1 bar, respectively. The production run, performed under the previously equilibrated conditions, consisted of 100 ns. During the equilibration and production processes an integration time step of 2 fs was employed. Energies and compressed coordinates were saved every 100 ps.

### 2.2. Structure and Trajectory Analysis

#### 2.2.1. Statistical Analysis

From the trajectories, the root-mean-square deviation (RMSD), root-mean-square fluctuation (RMSF) and hydrogen bonds were analyzed. RMSD and RMSF plots were constructed for each hACE-2 and Spike protein (wild type and gamma variant) separately. The hydrogen bond, RMSD and RMSF analyses, together with molecular geometric features, were extracted from the trajectories employing the Gromacs toolbox. The trajectories were visualized and analyzed with VMD v.1.9.4a51 and PyMOL [23,26].

#### 2.2.2. Molecular Mechanics Poison-Boltzmann Surface Area (MM-PBSA)

The MM-PBSA analysis was performed with *g_mmpbsa* program to calculate the binding affinity and the energy contributions of the residues to the protein-protein interaction [27,28]. The calculation procedure of energy terms consisted of three steps: calculation of the potential energy in vacuum, calculation of the polar solvation energy, and the calculation of the non-polar solvation energy. The electrostatic energy was computed with the addition of an exponential damping factor according to the Debye–Huckel theory [29]. This procedure in known as screened MM-PBSA. The average binding energy and the summary of the energy terms were obtained with the script MmPbSaStat.py [27]. In addition, the average of the energy contribution of each residue was obtained by using the script MmPbSaDecomp.py [27]. To solve the linear approximation of the Poisson-Boltzmann (PB) equation, the dielectric constants for solute and solvent were set to 2 and 80, respectively. An ionic strength was set to 150 mM. The apolar contribution was computed using a surface area approximation with *γ* equal to 0.0226778 kJmol^−1^ Å^−2^ and *b* (fitting parameter) equal to 3.84982 kJmol^−1^. A total of 100 frames, from each simulated complex, were employed.

#### 2.2.3. Entropy Calculations

The entropy calculations were performed employing the interaction entropy method developed by Zhang et al. [30,31]. This methodology provides a rigorous theoretical framework to compute the “gas-phase” component of the entropy change departing from fluctuations around the ensemble average of the interaction energy ∆*E^int^*:(1)$$-T\Delta S_{gas}=kTln\langle e^{\Delta E^{int}/kT}\rangle$$
with:(2)$$\langle e^{\Delta E^{int}/kT}\rangle =\frac{1}{N}\overset{N}{\underset{n}{\sum }}e^{\Delta E_{n}^{int}/kT}=\frac{1}{N}\overset{N}{\underset{n}{\sum }}e^{\left(E_{n}^{int}-\langle E^{int}\rangle \right)/kT}$$
where *T* and *k* correspond to temperature and Boltzmann constant, respectively. Brackets represents ensemble average over *N* simulation frames. Consequently, the term ∆*E_n_^in^*^t^ captures the deviation in the protein-protein interaction energy of frame *n* with respect to ensemble average. This methodology has been successfully applied to achieve a better characterization of the binding affinities of protein-ligands and protein-protein complexes [32,33,34]. The relative entropy contribution of residue *k* to the interaction entropy is computed as:(3)$$\Delta S_{k}=\Delta S_{gas}-\Delta S_{gas,k}$$
where ∆*S_gas_,_k_* is obtained from:(4)$$-T\Delta S_{gas,k}=kTln\left(\overset{N}{\underset{n}{\sum }}\prod_{i\;\forall \;i\neq k}^{S}e^{(R_{ni}-\frac{1}{N}\sum_{n}^{N}R_{ni})/kT}\right)-kTln\left(N\right)$$
with *R_ni_* representing the interaction entropy contribution of the residue *i* in the frame *n*. Accordantly, *S* and *N* corresponds to the total number of residues and simulation frames, respectively. Derivation of the Equation (4) is provided in the Appendix B. ∆*S_k_* captures the entropy change those results from ignoring the contribution of residue *k* to the total interaction entropy term. We propose the use of residue relative entropy contribution, Equation (3), as a tool to identify the key residues involved in the interaction entropy assessment. It is important to emphasize that the validity of residue relative entropy contribution, proposed here, is limited within the theoretical foundations and restrictions that allows the derivation of Equations (1) and (2) [30,31].

In the present work, the interaction energy contributions were obtained from g-mmpbsa software [27]. Eight hundred frames, from the period 20–100 ns, were considered for the interaction entropy computation. The following expression was used to compare the residue relative entropy contributions of residue *k* between the wild type and gamma complexes:(5)$$T\Delta \Delta S_{k}=T\Delta S_{k,gamma}-T\Delta S_{k,wt}$$

## 3. Results and Discussion

### 3.1. Molecular Dynamics Analysis

Molecular dynamic simulations were performed to characterize the structural features that determine the complex between hACE-2 and the spike protein. After 100 ns of simulation, the stability of both complexes, as a dimer, stabilizes at 15 ns, reaching a value between 2.5 and 3.5 Å (Figure 1A). For the hACE-2 receptor in both complexes, RMSD stabilizes at 15 ns to a value around the 2 Å (Figure 1B). Similar behavior is observed in the RMSD curves for the corresponding simulations of the spike protein (Figure 1C). Histograms were computed from each of the RMSD curves, discarding the first 20 ns, to confirm the presence of a gaussian-like distribution in the associated structural fluctuations (Figure 1). These results support the notion that after the first 20 ns the simulated systems reached an equilibrated state. Accordantly, all subsequent analyses were performed without considering the first 20 ns. Taken together, these results indicate the absence of large structural rearrangements and advocate for the formation of a stable complex.

A RMSF analysis was performed to gain further insights on the observed differential behavior at the hACE-2 receptor in the different complexes. Fluctuations in the hACE-2 receptor were more pronounced in the wild type complex than in its variant counterpart (Figure 2A). These differences are particularly notable at the regions comprised by segments L73–Y83, V93–K114, N134–Q139, N290–D299 and D335–Q340 (Figure 2B). In the spike protein of the gamma variant complex, fluctuations were smaller in residues A363–S375 (Figure 2B,C). Interestingly, most of these segments, with L73–Y83 being the exception, are not located at the complex binding interface. Thereby, the absence of major structural differences between wild type and gamma variant complexes suggests that differences observed experimentally should arise from the associated specific intermolecular interactions.

### 3.2. Effect of Mutations on the hACE-RBD Interaction

In the complex with the wild type protein, the formation of a salt bridge between K417, from the spike protein RBD, and the oxygens of the D30, from hACE2, was observed throughout the 80 ns (Figure 3A,B). The geometric features the characterize the K417 and D30 interaction are in agreement with the patterns observed in crystallographic data surveys [35]. Considering this geometric description, the salt bridge formed between K147 and D30 shows a prevalence of 82% of the analyzed time. This interaction is absent in the gamma variant complex, since the distance between the oxygens of the carboxylic acid of D30 and hydroxyl group of T417 is around 8 Å (Figure 3C). Consequently, these results, in agreement with previous studies [34], reinforce the idea that the K417T mutation should, on its own, imprint a negative effect on the complex binding affinity.

The N501Y mutation first appeared in the English variant, called the alpha variant, and resulted in an increased ability to evade humoral immunity [9]. In the literature, the effect of this mutation has been reported as an increment in the affinity for the receptor, therefore in a greater transmissibility of the virus [36]. In the wild type complex, the side chain N501 establishes hydrogen bonds with the side chains of residues Y41, K353 of the hACE2 receptor (Figure 4A). Nevertheless, its main polar interaction is established with the side chain of Q498; which is observed 98% of the time, within the spike protein RBD. Interestingly, these polar interactions are lost with the mutation N501Y. Considering that, with the incorporation of Y501, the binding partners of N501 may relocate their polar groups, it was decided to analyze the hydrogen bonds that the side chain of residues Y41 and K353 from hACE2 and Q498 from the spike protein establish in the gamma variant (Figure 4A). While the polar interaction of Y41 is basically unaltered upon the mutation at position 501 of the spike protein, K353 and Q498 suffered dramatic changes. Actually, the hydrogen bond interaction between K353 and Q498, with 87% prevalence in the wild type complex, is completely abolished with the mutation N501Y. Similarly, K353’s side chain interaction with D38 of hACE2 and G496 of the RBD are also drastically affected. In compensation, in the gamma complex, K353 interacts with the carbonyl oxygen of Y495 with a prevalence of 54%. In the case of Q498, its side chain polar interactions are practically abolished. For this important polar partner in the wild type complex, only low frequency interactions with residues V445, G446 and T500, within the RBD, and Q42 of the hACE2 receptor are observed in the gamma variant complex. Overall, in the gamma variant complex, at the immediate surroundings of position 501, not only are fewer hydrogen being formed, but they are also doing so with less frequency than those in the wild type complex (Figure 4A). Nevertheless, Y501 inserts itself into a pocket where it establishes a π-π stacking interaction with Y41 of hACE-2; an interaction that it is absent in the wild type complex (Figure 4B). This stacking interaction compensates, to some extent, for the lost hydrogen bonding partners that renders the mutation N501Y. Thereby, the gamma variant remains anchored strongly with hACE-2 despite not maintaining the hydrogen bonding network throughout the simulation.

Some of rearrangements described above, associated with N501Y, have been previously reported in the context of other variants and in the gamma variant itself [34,37,38,39,40]. It is accepted that this mutation, on its own, renders an increment in the binding affinity. This is explained as an energetic compensation of contributions from the hydrogen bond formed between N501 and K353, in the wild type, and that of the π-π interaction established between Y501 and Y41, in the variant complexes, along with a set of local structural rearrangements. However, within this established narrative, it has not been discussed that the hydrogen bond formed between the K353 and G496 in the wild type complex is substituted, in the variants, by the one established between the NH3+ group of K353 and the carbonyl oxygen of Y495 (Figure 4A). In order to this interaction to occur, the main chain of Y495 is required to adopt a *cis* configuration (Figure 5A). This main chain orientation is also observed in X-ray crystallographic data of several spike-hACE2 complexes (Figure 5B, Appendix A Appendix A) [21,41,42]. Interestingly, and in accordance with our simulations, the *cis* orientation is present in variants bearing the mutation N501Y; namely alpha, beta, gamma and omicron. Furthermore, the position 495 lays within the Loop 2 region in a more general context of coronaviruses [43]. Remarkably, this Loop 2 region was found to be essential in the recognition and binding of S proteins to hACE2. Taken together, these results establish the hypothesis that the main chain *cis* configuration of Y495 is promoted by the presence of tyrosine at the position 501 and, with it, its ability to bind to hACE2 is enhanced. This opens new routes to investigate the effects associated with the N501Y mutations that extend beyond the π-π interaction with Y41.

In the literature, it has been suggested that the E484K mutation confers to the virus the ability to evade the humoral immune system rather than imprinting a higher affinity for the receptor [19]. The region comprised between residues 470 and 490, bearing the E484K of the spike protein, is positioned near to a predominantly negatively charged region of hACE-2. In this context, the E484, in the wild type complex, faces an electrostatic opposition from residues E35 and E75 (Figure 6A). In contrast, the K484 of the gamma variant provides a better charge complementary at the interface with hACE-2 (Figure 6B). Nevertheless, neither in the wild type complex nor in the gamma variant complex, the residue at position 484 of the spike protein established a direct short-range interaction with specific residues of hACE-2. Thus, these observations suggest that the mutation E484K endows the gamma variant with a higher affinity for hACE-2, counteracting the effect of the K417T mutation.

The results presented above provide dynamic insights on the interactions of these mutations. On the simulation, the N501Y mutation rearranges the hydrogen bond network surrounding residues without compromising its enthalpic contribution. Regarding the E484K mutation, the results suggest a long-range electrostatic stabilization rather than a residue specific interaction. In contrast, the effect of the K417T mutation is much more localized since it determines the loss of a salt bridge with D30. Binding affinity estimations were performed to confirm these effects.

### 3.3. Effect of Mutations on Binding Affinity

In previous studies involving the wild type protein and the alpha variant of SARS-CoV-2 as well as SARS-CoV, the binding affinity values have been reported using the molecular mechanics-generalized born surface area (MM-GBSA) and MM-PBSA techniques [34,44,45]. Here, a screened version of MM-PBSA was performed in both protein complexes to assess the impact that each of the gamma variant mutations has on the binding energy. While the wild-type complex ∆H was estimated to be −55.88 ± 22.15 kcal/mol, the gamma variant complex reached an ∆H of −63.37 ± 20.59 kcal/mol (Table 1). Consequently, ∆∆H between the two complexes is expected to be around −7.49 kcal/mol, which dominates the binding process, since the −TΔΔS corresponds to −3.16 kcal/mol. Interestingly, the changes in the entropy suggest that complex formation in the gamma variant case is less penalized than in its wild type counterpart. It is important to consider that the MM-PBSA technique and interaction entropy overestimates by two to five times the delta G with respect to the experimental value [27]. With this in mind, the estimated change of ∆∆G −10.67 kcal/mol is in relatively good alignment with the reported experimental affinities (Kd_wt_= 26.34 ± 1.10 nM and Kd_gamma_= 5.16 ± 0.04 nM) [19]. A detailed observation of the free energy terms reveals that electrostatic and polar contributions condensate the main differences between the wild and gamma variant complexes (Table 1). In both cases the gamma variant displays favorable modifications according to the MM-PBSA approximation (∆∆E_elec_ = 32.67 kcal/mol and ∆∆G_pol_ = −45.34 kcal/mol). Nonetheless, despite this slight gain in binding affinity, this difference is not large enough to claim an increment of the gamma variant transmissibility.

A closer inspection on the specific contributions of each of the three mutations in the RBM indicate that the highest binding energy contribution variation was observed in the E484K mutation (Table 2). The computed change corresponds to ∆∆H = −1.79 kcal/mol for K417T and ∆∆H = −3.06 kcal/mol for E484K, respectively (Table 2). The mutation N501Y showed a slightly lower modification in its contributions to the binding affinity (∆∆H=1.17 kcal/mol). On the other hand, the specific residue contribution to the interaction entropy presented the residues S19, D30 and E35 in the hACE2 receptor as more affected by the mutations (Table 3). While the entropy changes on residues D30 and K31 can be associated with the K417T mutant, those observed at S19 cannot be attributed directly to any of the mutations. The computation of residue relative interaction contribution allows the assessment of the degree of participation of each residue in the binding affinity (Table 4, Appendix A Appendix A). As expected, the ∆∆G linked to mutations K417T (∆∆G = 2.32 kcal/mol) and E484K (∆∆G = −3.32 kcal/mol) has a compensatory character. Similarly, the mutation N501Y (∆∆G = 0.52 kcal/mol) shows a low net change per se. Taken together, while mutations K417T and E484K can be interpreted as intramolecular epistasis [46], the mutation N501Y apparently triggers the rearrangement of polar and apolar interactions.

Considering the above, the mutations may have little influence on the transmissibility of the virus. Nevertheless, these changes, as reviewed in the literature, have an influence concerning the interaction with neutralizing antibodies [18,47]. Specifically, in the gamma variant, a loss of affinity due to the E484K mutation has been observed with the monoclonal antibodies REGN10933 or Casirivimab (Regeneron) and Ly-CoV555 (Lilly). This loss of affinity is a consequence of the charge repulsion that occurs between lysine and the negatively charged hypervariable region of the antibodies [48]. Corresponding to the K417T mutation, K417 residue could form a salt bridge with E99 of the LY-CoV16 (Lilly) monoclonal antibody. However, this interaction is lost when the threonine takes its place in the gamma variant, resulting in a loss of binding affinity [49]. Moreover, the neutralizing antibodies COVOX-222 and EY6A were also tested to target the regions bearing mutations at residues 501 and 417. Surprisingly, the neutralization effect was unaltered [19]. Further analysis would be needed to describe the molecular elements that governs those interactions.

Regarding vaccines, these have shown efficacy against a large number of variants and today almost half of the world’s population, according to the World Health Organization (WHO), is vaccinated against COVID-19 [50,51,52]. However, new variants continue to appear, and others prevail due to lack of treatment and reinfection capacity. Thus, the gamma variant is no longer circulating, although it led the spread of the virus in South America during early 2021, according to GISAID and WHO data [53,54]. Despite this, we find it important due to the presence of compensatory mutations and the possible intramolecular epistasis that revolves around the evolution of SARS-CoV-2.

## 4. Conclusions

Viruses evolve to improve transmissibility, and become more lethal or evade the immune system to prevail over time. Specifically, with COVID-19, the lethality of the virus is low but its ability to transmit and reinfect seems to be improved with the appearance of the new variants. Compensatory mutations have been commonly observed in other circumstances with other viruses [46,47,48]. This work provides further evidence, based on free energy estimations, that supports the notion that the mutations K417T and E484K are compensatory. While keeping the balance on the binding affinity of the spike protein RBD towards the hACE2 receptor, they may also confer to the virus the ability to avoid the host immune system. However, the consequences associated with N501Y are more convoluted. Several research groups, employing distinct experimental and computational techniques, have reached the conclusion that the mutation N501Y confers to the virus a higher binding affinity for the hACE2 receptor [40]. Certainly, the main modification introduced by the tyrosine is the π-π interaction with Y41. Nonetheless, this mutation also disrupts the hydrogen bond between K353 of hACE2 and Q498. In principle, these losses and gains of interactions should imprint a near zero change in binding energy associated with mutation N501Y. Therefore, other consequences, additional to those already discussed in the literature, should be associated with this mutation. Our results allow us to identify the presence of ***cis*** configuration in the backbone of Y495, which allows the hydrogen bond interaction with the NH3+ group of K353. This ***cis*** configuration correlates with the presence of Y501 in X-ray crystallographic models of several variants (including alpha, beta, gamma and omicron). These structural modifications may also be present at Y495 in other coronaviruses [43]. Further analysis is required to completely unveil the consequences associated with the N501Y mutation.

## Figures and Tables

**Figure 1 molecules-27-02370-f001:**
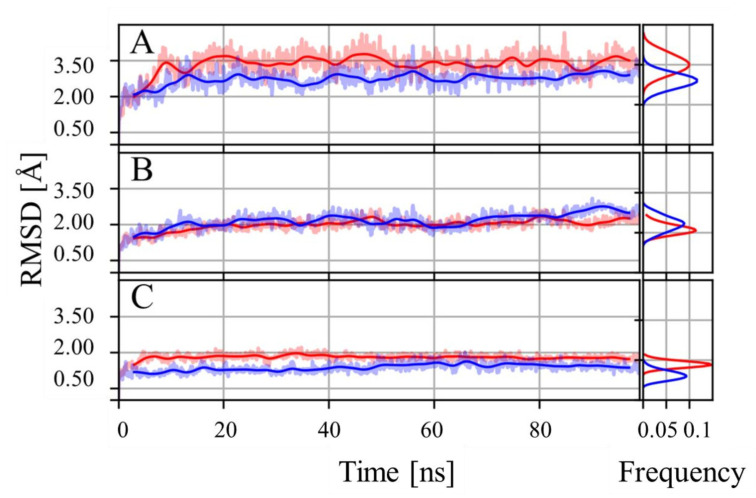
RMSD throughout 100 ns of MD simulation. Alpha carbon RMSD computed over the whole complex (**A**), hACE-2 receptor (**B**) and RBD of spike protein (**C**) from the wild type (blue) and gamma variant (red) simulated complexes. Bold continuous lines represent the weighted moving average over the original RMSD data (standard gaussian kernel with 30 frame window). The associated colored shadows correspond to the original RMSD data. RMSD histograms presented on the right end of each plot collect information starting at 20 ns.

**Figure 2 molecules-27-02370-f002:**
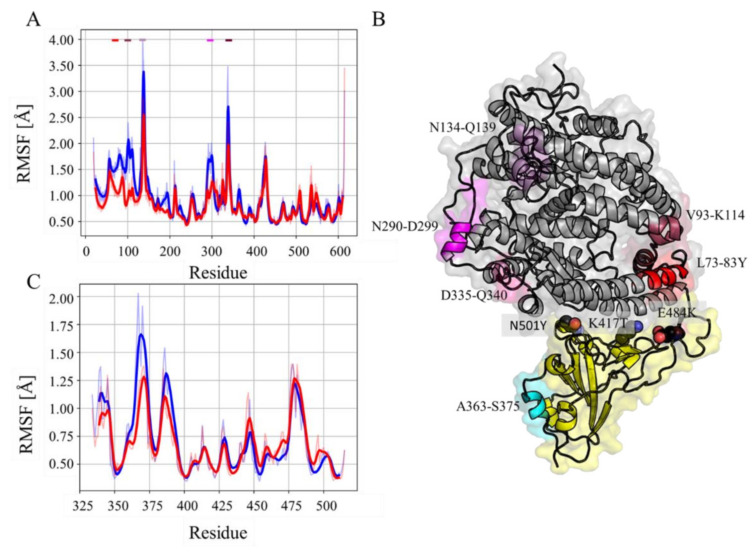
RMSF of bound proteins. Alpha carbon RMSF computed over the whole hACE-2 receptor (**A**) and RBD of spike protein (**C**) from the wild type (blue) and gamma variant (red) simulated complexes. Bold continuous lines represent weighted moving average over the original RMSF data (standard gaussian kernel with 5 residue window). The associated colored shadows correspond to the original RMSF data. The colored bars on the top each panel highlight regions with differential behavior between the complexes. These segments are mapped on the structure with the same color code (**B**). Residues bearing the key mutations of the RDB are shown in black sticks.

**Figure 3 molecules-27-02370-f003:**
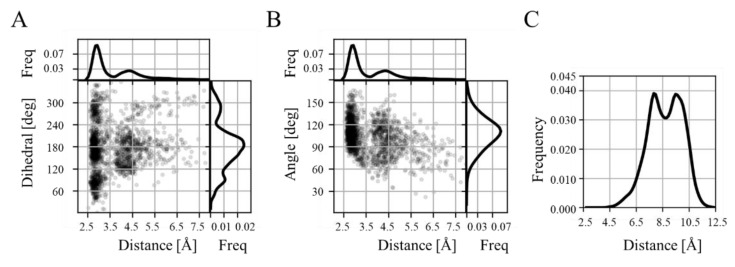
Structural highlights of the K417T mutation. (**A**) Distance-dihedral chart of the interaction of K417 with D30. Each dot in the plot represents the distance between K417N_ζ_ and D30O_δ_ and the dihedral form between C_δ_-C_ε_-N_ζ_ of K417 and O_δ_ of D30. (**B**) Distance-angle chart of the interaction of K417 with D30. Each dot in the plot represents the distance, as described for (**A**), and the angle form between C_ε_-N_ζ_ of K417 and O_δ_ of D30. Histograms for each single variable are presented at the edge of the chart in (**A**) and (**B**). (**C**) Histogram describing the distance between T417O_γ_ and D30O_δ_.

**Figure 4 molecules-27-02370-f004:**
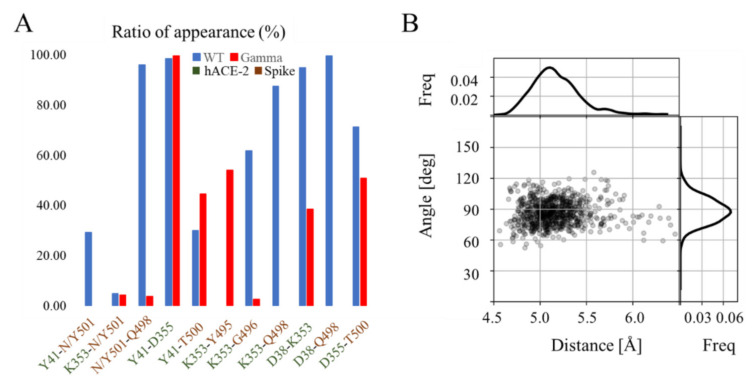
Structural highlights of the N501Y mutation. (**A**) Ratio of appearance (%) of the hydrogen binding partners with N(Y)501 residue throughout the last 80 ns of simulation for the wild type complex (blue) and gamma variant complex (red)). (**B**) Distance-angle chart of the π-π interaction between Y501 and Y41. Each dot in the plot represents the geometric center distance and normal vectors angle between the corresponding aromatic groups. Histograms for each single variable are presented at the edge of the chart.

**Figure 5 molecules-27-02370-f005:**
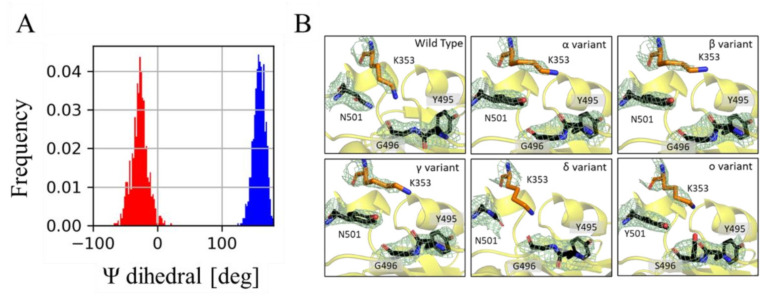
Mutation N501Y induces a cis configuration at 495Y main chain. (**A**) Histograms of ψ dihedral angle of Y495 from the wt (blue) and gamma variant (red) MD simulations. (**B**) 2Fo-Fc electron density maps around Y495 for the wt and VoC crystallographic complexes (PDB codes: 6MOJ, 7EKF, 7EKG, 7EKC, 7WBQ and 7WBP). The electron density, contoured at 1.5σ (pale green mesh), is shown around residues Y495, G(S)496 and N(Y)501 of the spike protein (black sticks), and residue K353 of the hACE2 receptor (orange sticks).

**Figure 6 molecules-27-02370-f006:**
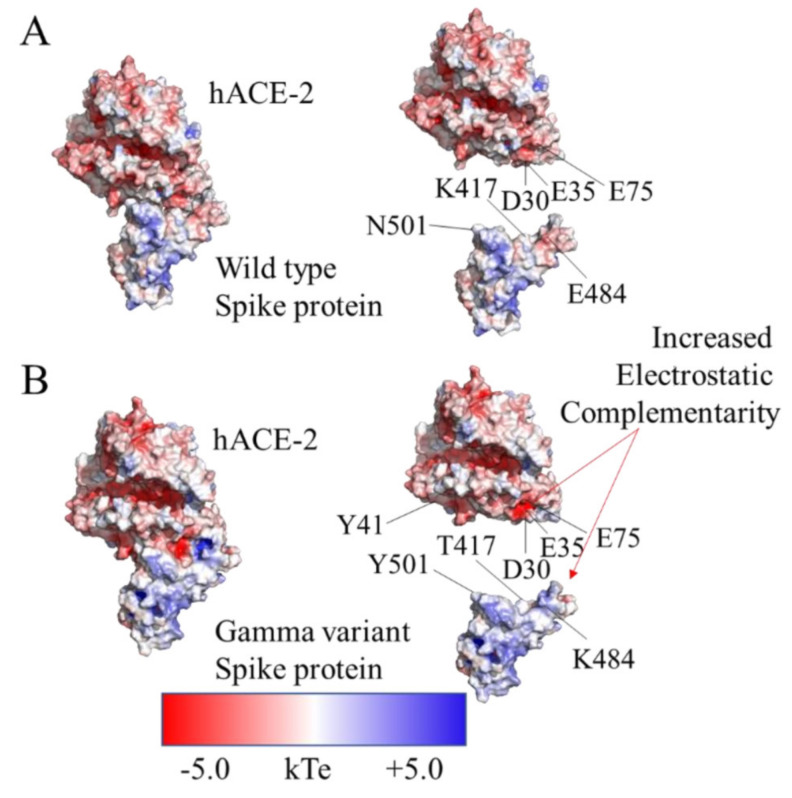
Structural highlights of the E484K mutation. APBS electrostatics analysis for both complexes: (**A**) wild type, and (**B**) gamma variant. Electrostatic complementarity due to the 484K mutation with residues E35 and E75 of hACE-2 are highlighted.

**Table 1 molecules-27-02370-t001:** MM-PBSA summary results for SARS-CoV-2 Spike RBD/hACE2 wild type complex and gamma variant complex. Values correspond to mean and standard error.

	Wild Type [kcal/mol]	Gamma Variant [kcal/mol]
∆E_elec_	−102.84 ± 13.36	−70.17 ± 8.50
∆G_non pol_	−11.92 ± 0.79	−10.99 ± 0.85
∆G_pol_	159.86 ± 19.70	114.52 ± 21.11
∆E_VdW_	−100.98 ± 6.16	−96.73 ± 5.59
∆H	−55.88 ± 22.15	−63.37 ± 20.59
−T∆S	29.45 ± 0.56	26.27 ± 1.47
∆G	−26.43 ± 24.68	−37.10 ± 23.50
∆G_exp_	−10.40	−11.39

**Table 2 molecules-27-02370-t002:** Major residue contributions to the binding affinity (∆H). Main residues contributing to binding affinity changes. The binding energy compensation is highlighted in red for the K417T, E484K and N501Y mutations. All values are in kcal/mol. Shaded cells correspond to hACE-2 residues.

WT			Gamma			∆∆H
Residue	∆H	Mean Std	Residue	∆H	Mean Std	
**S19**	3.60	0.47	**S19**	0.86	0.11	−2.74
**D30**	5.55	0.68	**D30**	1.33	0.16	−4.23
**K31**	3.89	0.50	**K31**	0.93	0.12	−2.96
**H34**	−2.93	0.12	**H34**	−0.70	0.03	2.23
**Y41**	−1.81	0.08	**Y41**	−0.43	0.02	1.38
**E329**	−1.79	0.08	**E329**	−0.43	0.02	1.36
**K353**	5.17	0.20	**K353**	1.24	0.05	−3.93
**D405**	2.43	0.16	**D405**	0.58	0.04	−1.85
**K417**	4.02	0.63	**T417**	0.96	0.15	−3.06
**L455**	−2.90	0.04	**L455**	−0.69	0.01	2.21
**F456**	−1.96	0.06	**F456**	−0.47	0.01	1.49
**E484**	2.35	0.24	**K484**	0.56	0.06	−1.79
**F486**	−2.55	0.11	**F486**	−0.61	0.03	1.94
**Q498**	−2.29	0.09	**Q498**	−0.55	0.02	1.74
**T500**	−1.66	0.10	**T500**	−0.40	0.02	1.27
**N501**	−1.54	0.11	**Y501**	−0.37	0.03	1.17
**Y505**	−3.63	0.09	**Y505**	−0.87	0.02	2.76

**Table 3 molecules-27-02370-t003:** Major residue contributions to the interaction entropy (−T∆S). Main residues contributing to binding affinity changes. The interaction entropy is highlighted in red for the K417T, E484K and N501Y mutations. All values are in kcal/mol. Shaded cells correspond to hACE-2 residues.

WT			Gamma			−T∆∆S
Residue	−T∆S	Mean Std	Residue	−T∆S	Mean Std	
**S19**	−0.86	0.16	**S19**	2.31	1.44	3.17
**D30**	−6.67	0.71	**D30**	−0.43	0.22	6.24
**K31**	0.01	0.97	**K31**	−2.74	0.89	−2.75
**H34**	1.63	0.79	**H34**	−0.24	0.63	−1.87
**E35**	0.70	0.25	**E35**	−2.29	1.37	−3.00
**D38**	0.49	0.64	**D38**	−1.80	0.97	−2.29
**K353**	0.49	0.34	**K353**	0.72	0.98	0.23
**K417**	−5.56	2.85	**T417**	−0.18	0.08	5.39
**E484**	−0.57	0.27	**K484**	−2.10	0.61	−1.53
**Q493**	0.08	0.12	**Q493**	−1.67	1.24	−1.75
**N501**	0.05	0.09	**Y501**	−0.59	0.31	−0.64

**Table 4 molecules-27-02370-t004:** Major residue contributions to the free energy (∆G). Free energy changes with a cut-off −2≤ and ≥2 with the exception of N501Y mutation. The free energy compensation is highlighted in red for the K417T, E484K and N501Y mutations. All values are in kcal/mol. Shaded cells correspond to hACE-2 residues.

	−T∆S		−T∆∆S	∆H		∆∆H	∆G		∆∆G
	WT	Gamma		WT	Gamma		WT	Gamma	
**D30**	−6.67	−0.43	6.24	5.55	1.33	−4.23	−1.12	0.89	2.01
**K31**	0.01	−2.74	−2.75	3.89	0.93	−2.96	3.90	−1.81	−5.71
**E35**	0.70	−2.29	−3.00	0.61	0.15	−0.46	1.31	−2.15	−3.46
**D38**	0.49	−1.80	−2.29	−0.05	−0.01	0.04	0.44	−1.81	−2.25
**E75**	−0.21	−1.43	−1.22	1.02	0.24	−0.78	0.82	−1.19	−2.00
**E329**	0.17	0.90	0.73	−1.79	−0.43	1.36	−1.62	0.47	2.09
**K353**	0.49	0.72	0.23	5.17	1.24	−3.93	5.66	1.95	−3.71
**D405**	1.04	0.21	−0.83	2.43	0.58	−1.85	3.47	0.79	−2.68
**K417T**	−5.56	−0.18	5.39	4.02	0.96	−3.06	−1.54	0.78	2.32
**F456**	−0.91	−0.06	0.84	−1.96	−0.47	1.49	−2.87	−0.53	2.34
**E484K**	−0.57	−2.10	−1.53	2.35	0.56	−1.79	1.78	−1.54	−3.32
**T500**	−0.75	0.35	1.10	−1.66	−0.40	1.27	−2.42	−0.05	2.37
**N501Y**	0.05	−0.59	−0.64	−1.54	−0.37	1.17	−1.48	−0.96	0.52
**Y505**	−0.33	−0.15	0.17	−3.63	−0.87	2.76	−3.96	−1.02	2.94

## Data Availability

MD trajectories can be found at https://drive.google.com/drive/folders/1qBof9FMPQwkm0cneBVIcywy2zKzW3BR7?usp=sharing. The code to compute the Residue Relative Interaction Entropy Contribution from a residue decomposed interaction energy file generated with g_mmpbsa can be found at https://github.com/marciniega/rriec.

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
