# Peer review of "Molecular Dynamics and MM-PBSA Analysis of the SARS-CoV-2 Gamma Variant in Complex with the hACE-2 Receptor"

_molecules, 2022, doi:10.3390/molecules27072370_

Round 1
Reviewer 1 Report
Comments:
In the manuscript, an interesting study is presented, which contains a lot of information based on theoretical methods. However, there still exist some un The paper can be published after some major corrections.
Comments:
1. The authors explained, “Four different regions were identified: i) the upper region comprising the segments S109-D335 and C361-D615, ii) the middle region constituted by the segment P336-M360 (bearing the β3-β4 turn), iii) the lower or interacting region form by the segment S19-L100 (containing helix α1 and α2) and iv) a hinge region linking the upper and lower regions Q101-L108.” Probably it is worth indicating Figure 2 at this stage to see the structure of hACE-2 interfaced with spike protein. Moreover, these four regions are differently colored to distinguish them, though the authors partly colored them.
2. In Figure 1, please explain what does the bold line indicates, probably, the mean value of every author-defined period. Possibly RMSD of the total complex (Appendix-A) might be helpful for the readers to understand the stability of the complex as well. Please show in the main text.
3. In Figure 2, please specify the color bar below the RMSF values corresponding to the color in Fig 2-B.
4. Since the distance indicated in Figures 3 A and B are too long for the salt bridge the reviewers request the reinvestigation of them in detail by showing the definition of the salt bridge in view of those in the stable structure. To judge the salt bridge, please use some plugin installed in Pymol or VMD and show the ratio (%) during the simulation.
5. In Figure 4 (D) the authors showed the hydrogen bonds in broken lines. Are they really included in the range of the typical hydrogen bonds? Please confirm them throughout. For this figure, please indicate the single letter AA and the corresponding number for the important amino acid residue
6. From Figures 4(A) and (D), hydrogen bonds formed tentatively. Are they really responsible for the stabilization? Please confirm them and show the ratios for each hydrogen bond.
7. Please erase thin lines around words A and B of Figure 5. Also please make the figures in the color bar clearer.
8. The authors excused that MM/GBSA or PBSA overestimate the binding free energy, but the “∆∆Gbind of -83.87 kcal/mol found by the MM-PBSA analysis is in relatively good agreement with the reported experimental affinities“. How did the author judge the theoretical results to explain the experimental values? Please explain in detail by showing what kind of formulae are used in converting the theoretical results into the corresponding experimental ones.
9. Judging from Table 1, the main cause in the mutation Gamma Variant comes from Delta G_pol, i.e., about 60 kcal/mol decrease from WT. Please explain in relation to the amino acid residues in the mutations of the gamma variant. This means that Appendix B should be discussed in detail.
10. In Figure 6, please add the single letters in RBD spike protein like N501Y, E484K, and K417T.
Reviewer 2 Report
The authors performed molecular dynamics simulations and MM-PBSA calculations to study the SARS-CoV-2 gamma variant and the influence of mutations on the interaction with the hACE-2 receptor. This topic is of great concern and interest. However, the studied variant has been discovered since early 2021. I recommend the authors provide more discussions about the necessity of choosing this gamma variant before this manuscript being accepted.
The following are comments:
- As I mentioned above, gamma variant has been reported for a year. The most concern SARS-CoV-2 variants are delta and omicron. Besides, the number of new detected sequences of gamma variant is closed to zero in October 2021 (see https://cov-lineages.org/global_report_P.1.html for details). Therefore, I strongly recommend the authors address their reason about choosing gamma variant but not delta or omicron.
- Figure 1: the legend shows that A is the RMSD of the hACE2 receptor and B is the RMSD of the SARS-CoV-2 spike protein. Why L150-158 said A is the RMSD of hACE2 receptor and the SARS-CoV-2 spike protein in the wild type complex? It is confused!
- Figure 1: the RMSD of gamma variant (blue) is not stabilized, which means the structure is not stable during the whole simulation.
- Is it feasible to further consider the contribution of entropy? whether the results will change?
- I searched in google scholar for the studies of gamma variant and found a series of papers (i.e. 3389/fphar.2021.717757, 10.1016/j.csbj.2021.07.026, 10.1038/s41467-021-27325-1, 10.1002/jcb.30142, acs.jctc.1c00965). They all investigated the binding affinity of gamma variant and studied the influence of mutations. What the innovativeness of this manuscript compared with these previous works?
Round 2
Reviewer 1 Report
In the revised manuscript, the authors have revised correctly according to the comments by reviewers. So the revised version is now ready for publishing.
Author Response
There was no comments from reviewer 1.
Reviewer 2 Report
The authors revised the manuscript and added entropy contribution. However, I’m still not satisfied.
- The authors relied that gamma variant can still be use as a mean to understand the consequences of their mutations, particularly, N501. As far as I know, N501 mutation has been widely studied and reported (such as papers listed in comment 5 of v1). They still haven’t provided enough reasons to explain why they chose gamma over the most concern variants (e.g. delta, omicron). Delta and omicron variants were also reported for long time ago. Besides, they didn’t provide their discussion in the manuscript.
- Why the RMSDs (Figure 1B-C in manuscript v2) are so different with that in manuscript v1 (Figure 1A-B)? For example, the RMSD in Figure 1B (V2) is around 2.0 for both blue and red lines, but they are 2.5 and 2.0 in Figure 1A (V1). They drew it wrong in the revised version or first version? These two figures do not correspond to each other? Or they modified the data of RMSD?
- They said new simulations were performed. What are the differences between new and old simulations? Why the system was not stable in old simulations but they become stable in new ones?
- It is good.
- The author did not answer the question positively. manuscript 10.1016/j.csbj.2021.07.026 corresponds to a study that does not involve gamma variant, but alpha and beta. This manuscript also includes analysis of the consequences of residue mutations, like N501. manuscript 10.1038/s41467-021-27325-1 reported how these mutations quantitatively affected the kinetic, thermodynamic and structural properties of RBD—ACE2 complex formation (include gamma variant). Manuscript acs.jctc.1c00965 also studied gamma variant and discussed its mutations. So, what’s the main advance of this manuscript? only free energy calculation?
- Besides, I find another problem. The authors reported the binding energies and free energies in the range of -661 to -886 kcal/mol. However, I read some related papers, and found that the binding energies reported in other papers were significantly smaller than that in this manuscript. For example, -57.81 kcal/mol in 1039/D1CP01611C, -12 to -17 kcal/mol in 10.1016/j.csbj.2021.07.026, -4.5 to -25.27 kcal/mol in 10.1021/acs.jcim.1c00241. As the authors written, the experimental binding energy are -10.4 and -11.39 kcal/mol. The question is why the authors got such strange values? More discussion is required.
